# Prediction of Structural Performance of Vinyl Ester Polymer Concrete Using FEM Elasto-Plastic Model

**DOI:** 10.3390/ma13184034

**Published:** 2020-09-11

**Authors:** Kazimierz Józefiak, Rafał Michalczyk

**Affiliations:** Department of Theoretical Mechanics, Pavement Mechanics and Railroad Engineering, Faculty of Civil Engineering, Warsaw University of Technology, Al. Armii Ludowej 16, PL 00-637 Warsaw, Poland

**Keywords:** polymer concrete, vinyl ester resin, durability, constitutive modeling, FEM modeling, concrete damaged plasticity

## Abstract

This paper presents the methodology for predicting the mechanical performance of structural elements made of polymer concrete (PC). A vinyl ester polymer concrete composition and the results of experimental studies to determine the basic mechanical properties of the material are presented. Following the strategy for sustainable development in the building industry, the material cost of polymer concrete was lowered by reducing the consumption of raw materials and the partial replacing of the microfiller fraction with recycled waste products—calcium fly ash. An accurate computational model enabling stress analysis is a convenient way to verify the suitability of PC as a construction material in structural applications. Due to difficulty in deriving an accurate analytical formula, numerical approximation (finite element method) was used as a method for solving the problem. Constitutive modeling of PC is a very important aspect of the strength calculations and here it was done within the framework of elasto-plasticity. Numerical evaluation of the static bearing capacity of PC manhole covers is shown as an example of the proposed FEM methodology. The results of computer simulations were compared with laboratory tests. Finally, the adequacy of the numerical modeling for testing new construction and material improvements is discussed. The study showed that the concrete damaged plasticity material model can be effectively used for the description of PC mechanical behavior.

## 1. Introduction

The use of polymeric materials in concrete formulations is well known, but in recent years the applications of polymer concrete (PC) have been broadened [1]. From the commonly used bonding agents for concrete surface repair of existing structures it now becomes a main construction material for non-structural, as well as structural, elements. Precast polymer concrete has been one of the most promising innovations, but its full potential is still to be reached. Application examples include precast polymer concrete, such as for drains, tanks, manholes, sewer pipes or lift stations [2], railway sleepers and structural beams [3]. Many other applications should be explored to take advantage of the excellent chemical resistance and improved physical properties such as high strength and impact resistance. Moreover, the ability to form intricate shapes and textures, excellent durability and ability to provide color can make them even more popular with architects.

Another potential application, which is a fast-emerging technology and a source of debate, is the use of PC for 3D printing in the construction industry. The use of polymer concrete in that context, contrary to cementitious-based concrete types, would provide a finished material that would be more impermeable, stronger, more durable and cure much more quickly [4].

Three principal classes of polymer concrete materials exist: (a) polymer-cement concrete (PCC), (b) polymer-impregnated concrete (PIC) and (c) polymer concrete (PC) [5]. The research presented in this paper is focused entirely on the latter.

Polymer concrete is a mixture of aggregates and resins or monomers that are used to replace Portland cement as a binder. The resultant polymer concrete mix hardens by polymerization after it is placed. Thermoset polymers are typically used as the matrix material, as the main advantage of these materials are low viscosity and the possibility of curing at room temperature or lower. Among the numerous thermoset polymers, unsaturated polyester (UP), vinyl ester (VE), methyl methacrylate (MMA), furan resin (FU) and epoxy resins have been applied in PC [6].

For example, orthophthalate unsaturated polyester resins combined with methyl ethyl ketone peroxide as an initiator are commonly used as binders [7,8]. Orak [9] analyzed the use of unsaturated isophthalic polyester resin and reported that it produced a hard, rigid polymer that can provide good mechanical properties. Mendis [10] conducted a study of commercial applications and property requirements for epoxies, which are another type of thermosetting polymer commonly used nowadays for polymer concrete. They have no volatiles, present different reactivity degrees depending on the curing agent and higher chemical resistance compared to polyesters. Further studies on epoxies with the addition of fiber reinforcement showed increased flexural strength and fracture properties [11]. Recently, Ferdous et al. [12] determined the optimal resin-to-filler and matrix-to-aggregate ratios of epoxy polymer concrete.

While epoxy resins are commonly used, much attention has been focused on the use of cheaper vinyl monomers such as polystyrene [13], methyl-methacrylite and furane derivative, usually in conjunction with a cross-linking agent. Czarnecki et al. [14] have studied the properties of PC based on low-viscosity vinyl ester resin. This type or resin is suitable where improved strength and chemical resistance is required.

The suitability of polymer concrete as the construction material in any structural application should be verified by the direct testing of the prototype. This allows for evaluating the final product, but not predicting or optimizing its features. The results are only valid for a given geometry and loading condition so cannot be easily generalized. Simultaneously, many fundamental studies on the mechanical properties of polymer concrete are available. These studies on the strength of polymer concrete under uniaxial tension–compression or more complex stress conditions can be used to predict the performance of any structure. An appropriate numerical model based on experimental tests can be used to design and evaluate the mechanical performance of any structural elements made of polymer concrete.

The objective of this study is to develop methodology for predicting the mechanical performance of structural elements made of polymer concrete. The derivation of analytical formulas is difficult since the behavior of this material is highly nonlinear. An accurate computational model is crucial in the design process and the stress analysis can be accomplished by means of finite element modeling [15]. An appropriate FEM model can indicate where areas of high stress and excessive deformation are expected and subsequently failure can be prevented by improving the geometrical shape or material. However, the literature barely mentions the constitutive modeling of polymer concrete, which is a very important aspect of any strength calculation. This paper presents how the concrete damaged plasticity (CDP) material model implemented in commercial Abaqus finite element method (FEM) software can be adopted for the description of polymer concrete. The proposed methodology of finding necessary input parameters of the numerical model is validated based on a laboratory test of a manhole cover. According the literature study carried out by the authors, the CDP model has not been previously used for polymer concrete.

The structure of the paper is as follows: in Section 2, the vinyl ester polymer concrete composition is presented. Next, experimental studies to determine the basic mechanical properties of the material are conducted. In Section 2.3, a finite element model is established based on the mechanical properties established in Section 2.2. Constitutive modeling of polymer concrete is done within the framework of elasto-plasticity. Additionally, a concept of a PC manhole cover is presented as an example. Next, the model of a PC manhole cover representing a bending test is developed.

In Section 3, results obtained from the FEM are described and modeling methodology is validated using a full-scale experiment of a manhole cover prototype bending to failure. In Section 4, a discussion of the results obtained is provided. Section 5 contains the final conclusions that summarize the most important achievements of the article.

## 2. Materials and Methods

### 2.1. Polymer Concrete Composition

#### 2.1.1. Vinyl-Ester Resin

The resin is the main binding material for the polymer concrete. The choice of resin was dictated by the operating conditions that the example product (manhole cover) has to withstand. As a result of the operation of solar radiation, the surface of the cover can heat up to about 60 °C. Unfortunately, this temperature is close to the polyester resin softening point. In contrast, vinyl ester resin has increased thermal resistance—it can be used up to a temperature of about 100 °C. For this reason, vinyl ester resin was chosen for the study. It is typically used when high durability, thermal stability and high corrosion resistance are needed. Vinyl ester, in comparison to polyester and epoxy resin, is in the middle of the performance spectrum. Vinyl ester, even if not as cheap as polyester, provides a lower cost than epoxy resin. It is made by the reaction between an epoxy resin and an unsaturated monocarboxylic acid. Essentially, it comprises a base of polyester resin strengthened with epoxy molecules. Vinyl ester has fewer open sites in its molecular chain. This causes some outstanding features like broad chemical resistance and low water penetration. It is more tolerant of stretching than polyesters. This makes it easier for it to absorb impact without damage. Additionally, it exhibits low peak exotherms and less shrinkage during curing [16]. For this experimental study, a trade name of Polimal VE-2MM produced by Ciech Resins (www.ciechresins.com) in the chemical plant Organika-Sarzyna, Poland was used. The properties of vinyl ester resin are summarized in Table 1. A composition of benzoyl peroxide with dimethylaniline and cobalt naphthenate was used as the initiator for vinyl ester (standard curing system for Polimal VE-2MM suggested by the manufacturer, see Table 2).

#### 2.1.2. Fine and Coarse Aggregates

The basic aggregate used to prepare the mix consisted of standard sand (Warsaw, Poland) and natural gravel. Commercially available standard sand (according to the European standard EN-196-1) with a maximum size of 2 mm was used. Choice of standard sand was dictated by the need for controlled grading of fine aggregate (the content of the smallest fractions was reduced in order to keep the amount of applied fluidized fly ash unchanged). The natural gravel of the fraction 2/4 mm was washed and dried to remove additional dust and ensure precise dosing of microfiller to the mix. Sand and gravel were used in a ratio of 1:2 (by mass).

#### 2.1.3. Microfiller

The addition of microfillers (e.g., quartz powder) in polymer concrete has been reported to enhance its mechanical properties and to improve the workability of the mix. Thanks to its spherical shape, fly ash grains may fill the empty voids between the larger grains of aggregate, reducing intermolecular friction and facilitating mixing.

A lot of effort is made to utilize wastes and by-products in civil industry as a part of a strategy for sustainable development. Following that trend, the material cost of polymer concrete can be lowered by reducing the consumption of raw materials and replacing it with less expensive equivalents. Partial replacing of the microfiller fraction with calcium fly ash—a product of combustion in fluidized bed boilers—can be an example of recycled waste products which, due to their properties, have not found a wide application. For this reason, the microfiller fraction consists of quartz powder and waste mineral powder. Quartz powder was prepared by grinding pure quartz sand (Warsaw, Poland). Fly ash applied in this polymer concrete mix was the by-product of the energy industry—the waste material of hard coal combustion from a cogeneration plant in Warsaw. Analysis of the microfillers’ (calcium fly ash quartz powder) size distributions was carried out at Warsaw University of Technology and the results can be found in [17]. The size of the particles of fly ash, as well as quartz, did not exceed 150 µm. The fly ash contained more fractions of grain sizes below 1 µm.

#### 2.1.4. Mix Design

The polymer concrete mix used in this research was designed according to a mathematical model. The model was elaborated during the project entitled “Polymer Concrete Composites” [18] and further developed in the following years at Warsaw University of Technology [17]. The fit of the mathematical equations in a regression process to the experimental data gives predictive power for material parameters that were not used in the original measurements, as long as these parameters are not too different from those that were measured. This was the method used in the abovementioned project, where laboratory tests were carried out using a statistical experimental design. The main advantage of using statistically designed experiments was the minimization of the number of the tests necessary to obtain the desired information on a material. As a result, a polynomial model was created—second degree polynomial functions of three variables—describing the relations between the analyzed properties and polymer concrete composition. This approach gives the opportunity to identify optimal compositions due to a particular property. It can be presented graphically (an example in Figure 1, discussed below). The chosen input variables were: x1 = A/B, x2 = B/M, x3 = P/M (relative ratios by mass where: A—aggregate, B—binder, M—microfiller, P—calcium fly ash). In general, it is convenient to characterize the composition of polymer concrete by these parameters used in an optimization approach:A/B [g/g]—aggregate to binder ratio by mass;B/M [g/g]—microfiller to binder ratio by mass;P/M [g/g]—waste powder to microfiller mass ratio.

The effect of silica microfiller replacement by calcium fly ash on selected polymer concrete mechanical properties (compressive, flexural and tensile strength), as well as binder hardening characteristics, were investigated in previous research [17,19]. The research concluded that the content of the calcium fly ash should not exceed 50% of total microfiller. A higher content caused a significant decrease in the workability (improper compaction of the mix) and, in consequence, mechanical properties decreased significantly.

The composition of vinyl ester concrete for manhole covers was selected using this previously developed mathematical model [17,18]. Considering the strength and workability of the mix, the main factors were aggregate to binder ratio by mass (A/B) and the degree of microfiller substitution with fly ash (P/M). The following ranges of values were considered: A/B = 5 ÷ 7, P/M = 0.3 ÷ 0.7. It was assumed that parameter B/M equaled 0.5 (half of the variability range) as is often recommended in the guidelines for designing the composition of polymer concretes. The composition of vinyl ester concrete was determined based on the technological tests (consistency of the mix appropriate for filling the mold): (A/B = 6.0; B/M = 0.5; P/M = 0.5). This composition is characterized by technical properties not worse than in the case of a regular (non-modified) one but, at the same time, it is cheaper. The mixing proportions by weight are presented in Table 3. The mechanical characteristics are illustrated in Figure 1. The plot presents the compressive/tensile strength of concrete mix depending on two material variables (A/B and P/M). The third variable was fixed (B/M = 0.5) as described earlier. Expected values of the strength properties are also marked on the plot.

### 2.2. Material Characterization

Mechanical properties are necessary to develop and calibrate a constitutive model of polymer concrete. For this reason, the next step was to test the material in compression, tension, flexure and fracture to obtain necessary parameters. Additionally, the homogeneity of different specimens was assessed.

The flexural strength under one-point loading was done in accordance with EN 12390-5. Beams of 40 × 40 × 160 mm were cast and cured in water for 28 days. The flexural strength test was carried out using the Instron 5567 testing machine (Instron, Norwood, MA, USA), as shown in Figure 2.

The tensile strength of the polymer concrete was experimentally measured using the Instron 5567 testing machine. Three specimens from the batch were tested in order to get the split tensile strength. The samples were demolded after 24 h of casting and allowed to cure for 7 days. The force at failure was recorded (see Figure 3).

Compressive strength test results are crucial in studying concrete, especially when considering FEM modeling for structural applications. The tests were undertaken in accordance with the technical requirements outlined in the EN 12390-3 standard using the hydraulic press Controls MCC8 (CONTROLS S.p.A, Liscate, Italy). Standard cubes (d = 100 mm) and cylinders (d = 100 mm and h = 200 mm) were molded using the designed polymer concrete mix. Additionally, for comparative purposes, two halves of each specimen after the flexural test were used to conduct a compression test. The compressive strength test setup is shown in Figure 4a.

The modulus of elasticity of polymer concrete was determined by conducting the laboratory test on cylindrical specimens. The compressive load was applied to the specimens in the longitudinal direction and the deformation of the specimen with respect to different load values was analyzed. Three strain gauges were attached to the concrete cylinder to measure the axial deformation during the compression test as shown in Figure 4c. Both initial and secant modulus of elasticity were calculated using recorded data.

To fully reveal the material characteristics on fracture toughness, a wedge splitting test (WST) was conducted as shown in Figure 4b. A WST was chosen for this purpose as a stable test, i.e., with a descending branch in the overall load-deformation diagram [20]. The aim of WST test was to measure the amount of energy necessary to split the specimen into two halves. This fracture energy, divided by the projected fracture area was, assumed to be the specific fracture energy G_F_. Three samples of 100 × 100 × 100 mm with an initial notch 30 mm wide and 20 mm thick were used. The test was undertaken with a controlled speed of notch opening of v_COD_ = 0.1 mm/min. The obtained graphs of the vertical load versus the crack mouth opened deflection (CMOD) are shown in Figure 5.

The mechanical properties of the PC are summarized in Table 4. These results will be used in the next section in order to identify input data for the FEM constitutive model.

### 2.3. Development of the Finite Element Model

#### 2.3.1. Concrete Damaged Plasticity Constitutive Model

The concrete damaged plasticity (CDP) constitutive model implemented in Abaqus software (version 6.16, Dassault Systèmes, Vélizy-Villacoublay, France) consists of the combination of non-associated multi-hardening plasticity and scalar (isotropic) damaged elasticity to describe the irreversible damage that occurs during the fracturing process [19]. The yield function of the CDP model in plane stress space is shown in Figure 6. A detailed description of the CDP model can be found in [21,22]. The CDP model was originally developed for the description of cement concrete. As it was pointed out earlier, according to the authors’ literature study, the CDP model has not been previously used for polymer concrete. Here, the model was adopted to numerically predict the response of a manhole cover made of vinyl ester polymer concrete. In this section, the procedure of identifying the constitutive model parameters will be briefly described. Necessary mechanical properties used in this process were established in Section 2 of the article by means of various laboratory tests (see Table 4).

In order to fully define the CDP model, one has to provide many parameters defining the elastic response, the shape of the yield surface and the plastic potential, as well as tabular data describing the concrete behavior in uniaxial compression and tension. Herein, all these parameters were assumed based on laboratory test results, the literature or suggested default values in Abaqus.

The Young’s modulus describing the material response in the elastic range was assumed to be equal to the mean Young’s modulus,
$E_{cm}$ de
$E_{cm}$
termined in the uniaxial compression test performed on cylindrical samples. A typical value of the Poisson’s ratio for concrete, 0.2, was assumed.

The parameters of the yield function and plastic potential are summarized in Table 5. The ratio of initial equibiaxial compressive yield stress to initial uniaxial compressive yield stress (see Figure 6), $f_{b0}/f_{c0}$ was $f_{b0}/f_{c0}$ as assumed to be 1.05, as for high-strength concrete [23,24,25]. The ratio of the second stress invariant on the tensile meridian to that on the compressive meridian, *K*, which K controls the shape of the CDP yield surface in the deviator plane, was assumed be to 0.667. The default value of the dilation angle ψ, ψ, was adopted. In the CDP model, the Drucker–Prager hyperbolic function is used for the flow potential function (non-associated flow). This generates another parameter, χ, referred to as eccentricity, that defines the rate at which the flow potential function approaches the asymptote. This parameter was determined as the ratio of mean uniaxial tensile strength to mean uniaxial compression strength.
(1)$$\varepsilon =\frac{f_{ctm}}{f_{cm}}=\frac{13.36MPa}{70.34Mpa}\approx 0.2$$

Compressive behavior of the material outside the elastic range was assumed according to the following relation between compressive stress $\sigma_{c}$ and compressive strain $\varepsilon_{c}$ [26]:(2)$$\sigma_{c}\left(\varepsilon_{c}\right)=f_{cm}\frac{k\eta }{1+\left(k-2\right)\eta }\;k=\frac{1.1E_{\mathrm{cm}}\varepsilon_{c1}}{f_{\mathrm{cm}}}\;\mathrm{for}\;0<\left|\varepsilon_{c}\right|{<\varepsilon }_{\mathrm{cu}1}$$
where $\eta =\varepsilon_{c}/\varepsilon_{c1}$, $f_{cm}$ is the mean value of the compressive strength, $\varepsilon_{c1}$ is the strain corresponding to $f_{cm}$, $E_{cm}$ is the mean Young’s modulus in compression and $\varepsilon_{cu1}$ is the maximum strain for which the above relation holds. Material parameters $f_{cm}$, $E_{cm}$, $\varepsilon_{c1}$ and $\varepsilon_{cu1}$ were estimated based on the uniaxial compression test results. Additionally, it was assumed that the end of the elastic range corresponded to 40% of the uniaxial compressive strength, i.e., $0.4f_{cm}$. For $\varepsilon_{c}>\varepsilon_{cu1}$ a close-to-linear drop of stress was assumed before failure. The resulting CDP model input is shown in Figure 7a. In this figure, the curve begins from the value of $0.4f_{cm}$ and its maximum corresponds to $f_{cm}$.

Tension stiffening data were defined in terms of the cracking strain, $\varepsilon_{t}^{ck}$. The curve, shown in Figure 8a, begins with the value of $\sigma_{t}=f_{ctm}$, followed by an exponential decrease to the point of $\sigma_{t}=0.01f_{ctm}$ and cracking strain $\varepsilon_{t}^{ck}=10\varepsilon_{t}^{el}\;$, where
(3)$$\varepsilon_{t}^{el}=\frac{f_{ctm}}{E}=\frac{13.36MPa}{21.802GPa}=0.061\%$$
after which a constant value of tension stress was assumed.

The CDP model assumes that the reduction of the elastic modulus is given in terms of a scalar degradation variable d, so that [21]
(4)$$E=\left(1-d\right)E_{0}$$
where $E_{0}=E_{cm}$ is the initial (undamaged) modulus of the material. The degradation variable is defined separately for compression ($d_{c}$) and tension ($d_{t}$) as the function of compressive inelastic strain ($\varepsilon_{c}^{in}$) and tensile cracking strain ($\varepsilon_{t}^{ck}$), respectively. Degradation parameters can be later used in postprocessing to visualize possible crack directions in a structure after failure. Plastic strain values are automatically calculated using the following formulas:(5)$$\varepsilon_{c}^{pl}=\varepsilon_{c}^{in}-\frac{d_{c}\sigma_{c}}{\left(1-d_{c}\right)E_{0}}\;,\varepsilon_{t}^{pl}=\varepsilon_{t}^{ck}-\frac{d_{t}\sigma_{t}}{\left(1-d_{t}\right)E_{0}}$$

The damage parameter was calculated both for compression and tension from the compression and tension behavior as the percentage drop of compressive or tensile strength, respectively. Additionally, care was taken to ensure that no negative plastic strain occurred according to Equation (5). The damage parameter input data are shown in Figure 7b and Figure 8b.

Finally, a small viscosity parameter was introduced in the CDP material to improve the numerical stability of calculations (see Table 5). To eliminate possible influence on the results, the internal energy of the model during analysis was compared with the case with no viscous stabilization.

#### 2.3.2. Finite Element Model of the Manhole Cover

The manhole cover was chosen as an example of using PC for the production of prefabricated elements [27]. This type of cover is usually made of ductile iron and for this reason they can be stolen, causing a potential threat to road users. Additionally, high chemical resistance and high mechanical strength recommend the use of PC as an alternative material.

The concrete manhole cover geometry was adopted based on typical cast-iron covers commonly used in Europe. The diameter of the plate was assumed to be 640 mm and its thickness to be 32 mm. The model geometry and boundary conditions were related to a laboratory test setup shown in Figure 9. The goal of the numerical analysis was to predict carrying capacity as well as the load versus deflection curve of the manhole cover made of vinyl ester polymer concrete. A calibrated and validated finite element method (FEM) model can serve as a convenient tool for the optimization and/or preliminary verification of various design approaches.

Axisymmetric and three-dimensional models were built. The axisymmetric model geometry is shown in Figure 10. In both models, the steel piston and base (see Figure 9) were modeled as analytical rigid surfaces which helped to decrease the computational cost of the analysis.

For the axisymmetric case, the concrete plate was discretized with four-node bilinear, reduced integration finite elements with hourglass control, which led to a fully regular rectangular mesh. The different average global size of these elements was assumed for convergence study. The three-dimensional plate was discretized using eight-node linear, reduced integration brick elements. The finite element mesh of the 3D model is shown in Figure 11, where elements with an aspect ratio greater than 1.5 are highlighted to ensure that there are no poor-quality elements in the zone where damage was expected.

Contact was modeled between steel elements (simplified to analytical rigid surfaces) and the concrete cover with a penalty constraint enforcement method. The basic Coulomb friction model was used to describe the frictional behavior with a coefficient of friction equal to 0.4.

The analysis was carried out in two steps. In the first step, gravity was applied, whereas in the second, the Static/Riks [21,28] (modified Riks algorithm) procedure, available in Abaqus, was used in order to analyze the non-linear response of the concrete manhole cover before and, to some extent, after failure. A large displacement formulation was used to account for any geometric nonlinearity.

## 3. Results

The mesh convergence study was a primary consideration. A sufficiently refined mesh, not overly demanding on computing resources, was used to ensure that the results from the FEM simulation were adequate. Subsequently, an FEM stress and strain analysis was carried out with cover behavior assessed in a static load test.

### 3.1. Mesh Convergence Test and Internal Energy Comparison

A mesh convergence test was carried out using the axisymmetric model. Curves of the load proportionality factor [21] versus deflection in the middle of the plate were obtained using the Riks algorithm for four different average global element sizes. The input force to the model was equal to 1 kN (see Figure 9a) so that the load proportionality factor could be read as the applied force in kN. The results of the analysis are shown in Figure 12. For element sizes smaller than 3 mm, a sufficient convergence was obtained.

In order to check if the viscous parameter μ (see Table 5) was small enough to eliminate any interference in the results, internal energy for the whole model with an average element size of 1 mm was compared to the case with no viscous stabilization. The comparison is shown in Figure 13.

### 3.2. Stress–Strain Analysis

Figure 14 shows a contour plot of the damage parameter $d_{t}$ after failure obtained from the axisymmetric model with a global mesh size of 0.5 mm. In Figure 15 and Figure 16, contour plots of the maximum principal logarithmic measure of strain and maximum principal stress are shown, respectively. The deformed configuration is shown with displacements scaled 10 times in order to present the response of the structure. As can be seen in Figure 14, the model predicts the localization of cracks quite well. However, the authors have not yet collected enough experimental data to deduce if the predicted crack directions are accurate. In Figure 16, in the failure region, the tension strength is exceeded due to the averaging and interpolation of stress values in the FEM mesh. The actual values of tension strength are not exceeded at stress integration points.

Figure 17 shows a comparison of force–deflection curves for 3D and axisymmetric models. The mesh of the 3D model had to be assumed to be coarser due to the higher computational cost. Results obtained from the 3D model are presented in Figure 18 and Figure 19. As can be seen, some mesh dependency is revealed in the 3D model of the plate in the form of rectangular strain localization zones. In the authors’ opinion, this mesh dependency contributes to the observed difference in force–deflection curves between the axisymmetric and 3D models. The 3D model served for verification, however, the axisymmetric idealization is preferred as a much finer mesh can be used.

The FE analysis results were compared with data obtained from the experiment presented in Figure 11. The localization of strain gauges used in the experiment is shown in Figure 20 in relation to the axisymmetric FEM model. The comparison of radial strain between the experiment and FEM model is shown in Figure 21.

## 4. Discussion

The results presented in this paper show the possibility for the usage of vinyl ester polymer concrete in the design of manhole covers. However, the structural capacity predicted by the model and experiment is still quite low and some reinforcement should be considered. For example, the introduction of rebar in the design could help to achieve a higher strength class, appropriate for heavy traffic applications. The created numerical model is also suited for the analysis of manhole covers subjected to more complex loading types, e.g., dynamic. On the other hand, it may be used for low-cost and faster verification of new designs—both geometrical and material changes are possible. Thus, some expensive real-scale experiments may be eliminated.

Further testing is needed in order to fully validate the possibility of using the concrete damaged plasticity model for the prediction of the behavior of vinyl ester polymer concrete. Nevertheless, the presented approach gives promising results and can be used for other types of PC precast elements.

## 5. Conclusions

This study presents the methodology for predicting the mechanical performance of structural elements made of polymer concrete (PC). In the paper, polymer concrete composition was described. The concrete material was tested in order to evaluate its applicability as a cheaper and more durable alternative for commonly used cast-iron covers. The laboratory test data was later used to find material parameters of the concrete damaged plasticity constitutive model. An FEM model of the concrete manhole cover was build according to the conditions of a laboratory test setup. The numerical model was used mainly to predict the carrying capacity of the concrete slab subjected to the loading generated by a steel piston of a hydraulic press moving vertically along the center of the concrete plate.

The main findings of this paper can be summarized in the following points:The concrete damage plasticity (CDP) material model can be successfully adopted to simulate the nonlinear mechanical behavior of polymer concrete.The CDP model takes many parameters and finding these parameters based on standard laboratory test data is not straightforward. The authors showed a clear procedure of finding CDP model input data based on standard laboratory tests.The CDP model was originally developed for the description of cement concrete. For polymer concrete, the authors proposed how to make necessary assumptions regarding the post-failure behavior.The authors showed that PC can be considered for the design of manhole covers. In this study, manhole cover made of plain PC showed too little structural capacity. However, in the authors’ opinion, the numerical approach presented here can still be considered as a valuable design tool (e.g., for manhole covers made of reinforced PC). The numerical solution can be used to choose the type and geometry of the reinforcement, which will be the subject of future studies. Evaluating the application of a certain material model and formulating necessarily steps for finding material parameters is an important milestone before proceeding to the reinforcement design, as it can significantly limit the number of laboratory tests.

## Figures and Tables

**Figure 1 materials-13-04034-f001:**
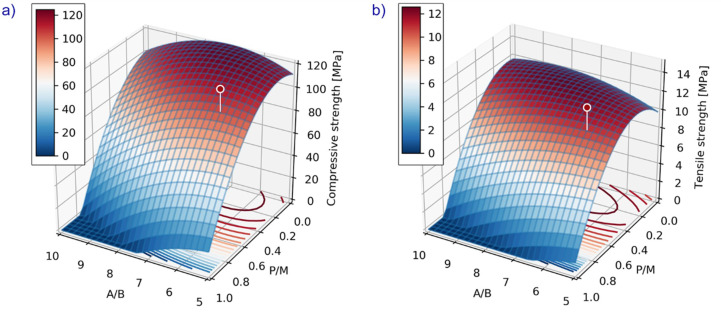
Compressive (**a**) and tensile (**b**) strength of concrete mix vs. the material variables x1 = A/B (aggregate to polymer binder mass ratio), x2 = P/M (waste powder to microfiller mass ratio), B/M = 0.5.

**Figure 2 materials-13-04034-f002:**
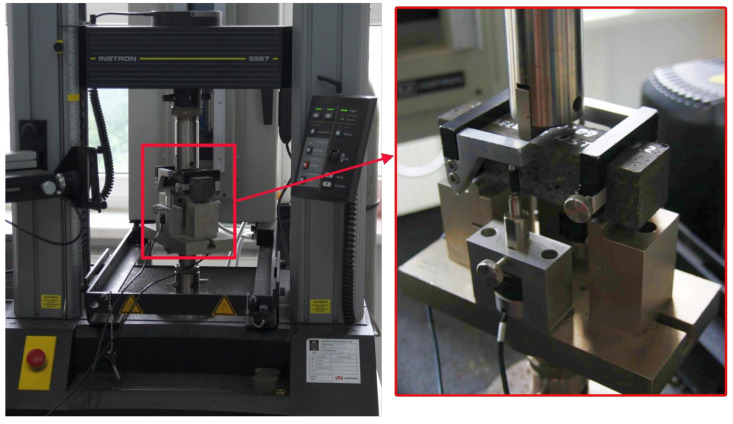
Bending strength test on Inston 5567 machine.

**Figure 3 materials-13-04034-f003:**
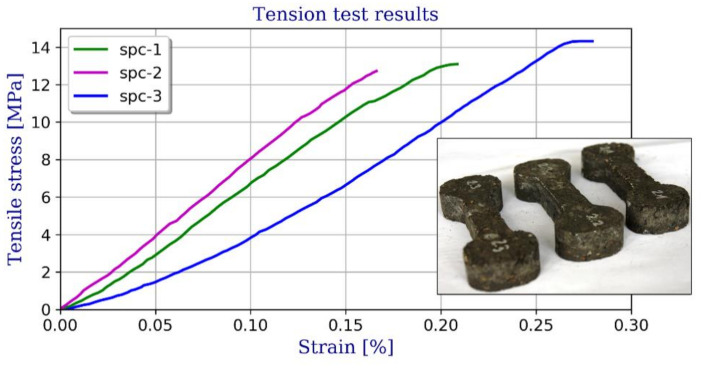
Stress–strain relationships for tension and specimens used for testing.

**Figure 4 materials-13-04034-f004:**
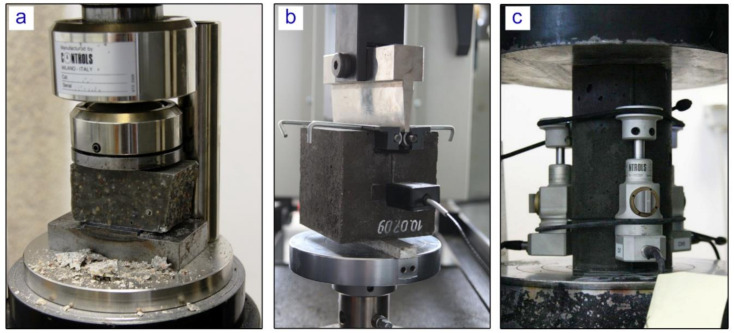
(**a**) Compressive strength test on Controls MCC8 hydraulic press, (**b**) wedge splitting test (WST) test on the Instron 5567 machine, (**c**) modulus of elasticity testing.

**Figure 5 materials-13-04034-f005:**
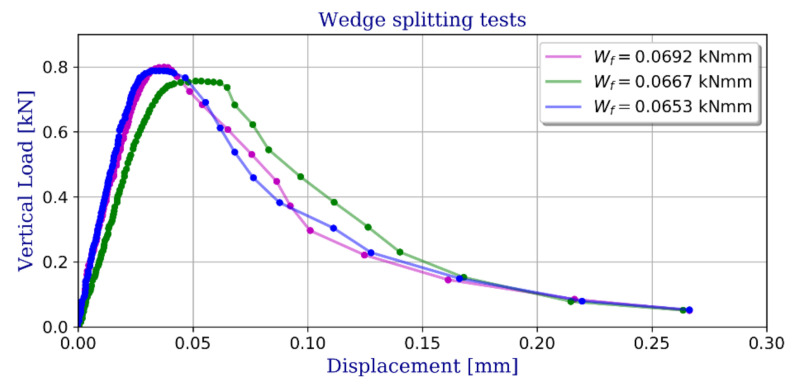
Wedge splitting tests—vertical load vs. crack mouth opened deflection (CMOD).

**Figure 6 materials-13-04034-f006:**
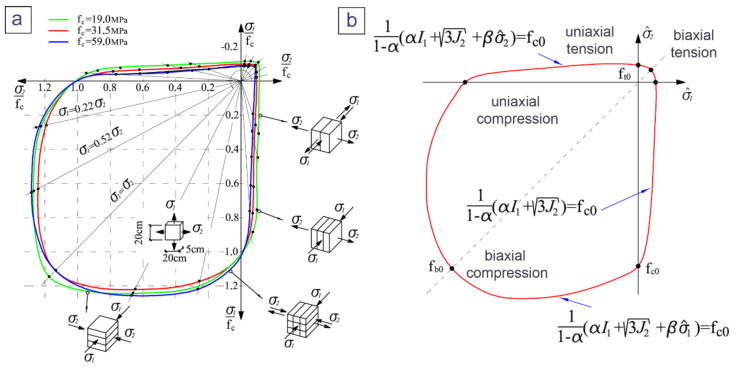
Biaxial strength of concrete according to [23] (**a**) and Lubliner [22] yield function in plane stress space (**b**).

**Figure 7 materials-13-04034-f007:**
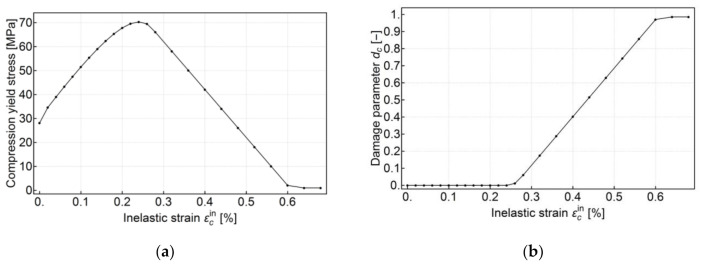
Compressive behavior assumed for the concrete damaged plasticity (CDP) model, (**a**) uniaxial compressive behavior outside the elastic range defined as a tabular function of inelastic (or crushing) strain $\varepsilon_{c}^{in}$ (**b**) uniaxial compression damage variable $d_{c}$ as a tabular function of inelastic strain.

**Figure 8 materials-13-04034-f008:**
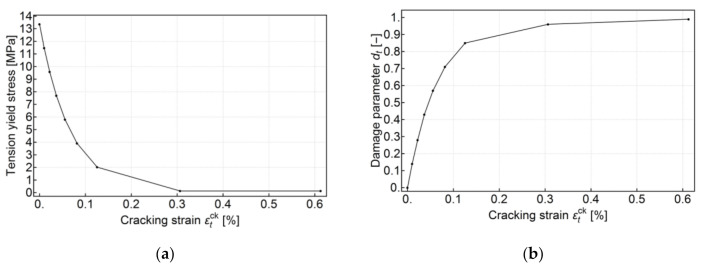
Tension behavior assumed for the CDP model, (**a**) post-failure stress as a function of cracking strain $\varepsilon_{t}^{ck}$, (**b**) uniaxial tension damage variable $d_{t}$ as a tabular function of cracking strain.

**Figure 9 materials-13-04034-f009:**
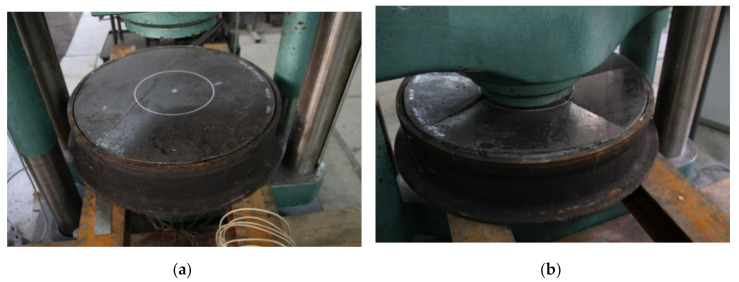
Laboratory test setup of a vinyl ester polymer concrete manhole cover [27], (**a**) before loading, (**b**) after failure.

**Figure 10 materials-13-04034-f010:**
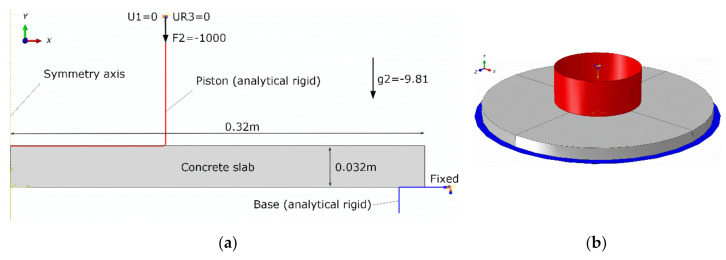
Geometry and boundary conditions (SI units) of the finite element model: (**a**) axisymmetric model, (**b**) 3D model view.

**Figure 11 materials-13-04034-f011:**
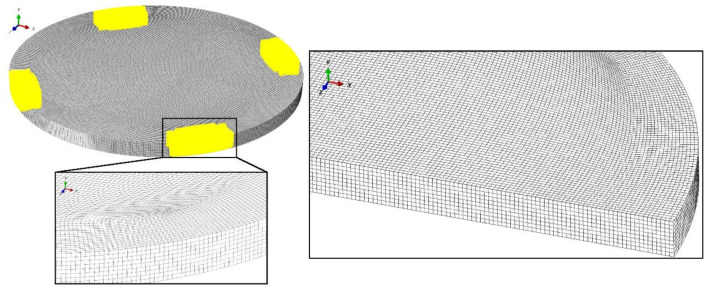
Finite element mesh of the 3D model with highlighted (yellow) elements with an aspect ratio greater than 1.5.

**Figure 12 materials-13-04034-f012:**
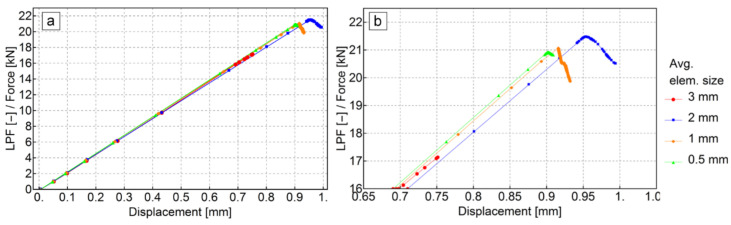
Load versus deflection curve comparison for different mesh densities of the axisymmetric model: (**a**) full structure response, (**b**) zoomed-in view of the failure region.

**Figure 13 materials-13-04034-f013:**
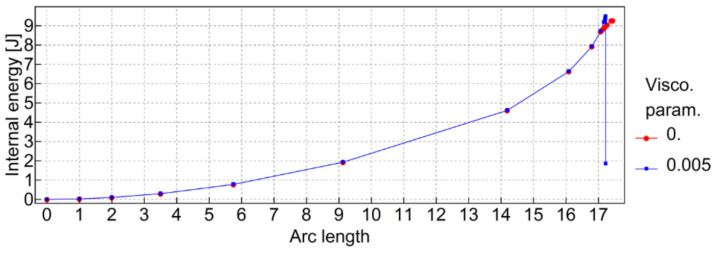
Internal energy comparison for the whole model with and without viscous stabilization.

**Figure 14 materials-13-04034-f014:**
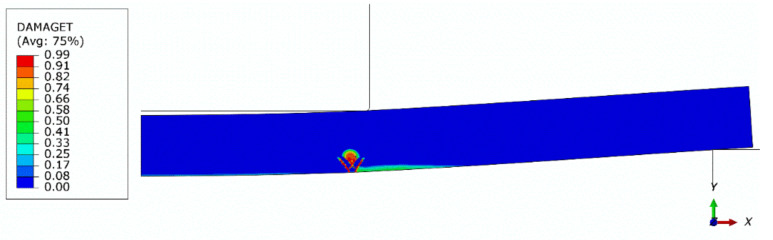
Contour plot of the tension damage parameter ($d_{t}$) after failure (axisymmetric model, global mesh size: 0.5 mm); deformed configuration; displacements scaled 10 times.

**Figure 15 materials-13-04034-f015:**
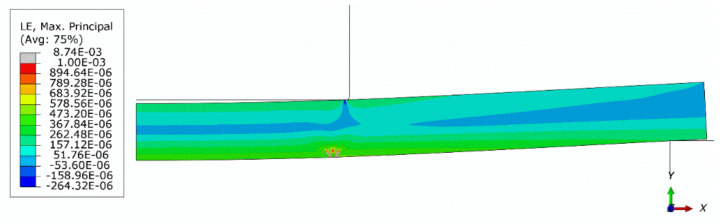
Contour plot of the maximum principal logarithmic measure of strain (axisymmetric model, global mesh size: 0.5 mm); deformed configuration; displacements scaled 10 times.

**Figure 16 materials-13-04034-f016:**
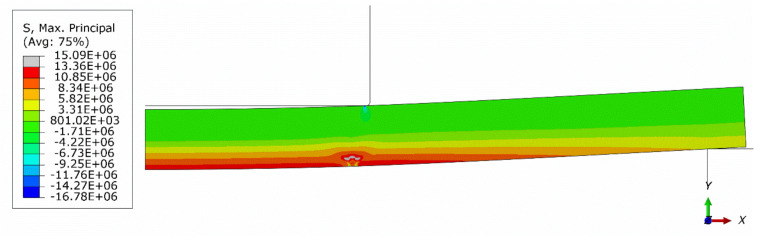
Contour plot of maximum principal stress after failure (axisymmetric model, global mesh size: 0.5 mm); deformed configuration; displacements scaled 10 times.

**Figure 17 materials-13-04034-f017:**
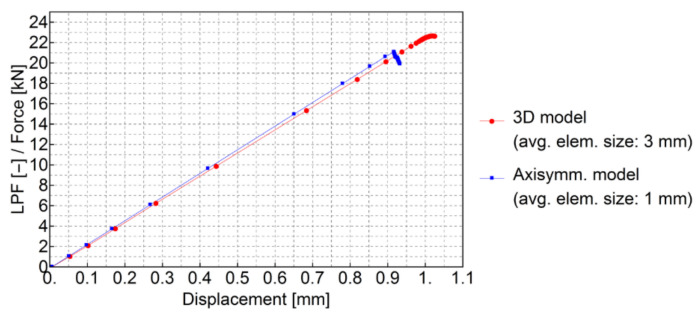
Comparison of force–deflection curves of 3D and axisymmetric models.

**Figure 18 materials-13-04034-f018:**
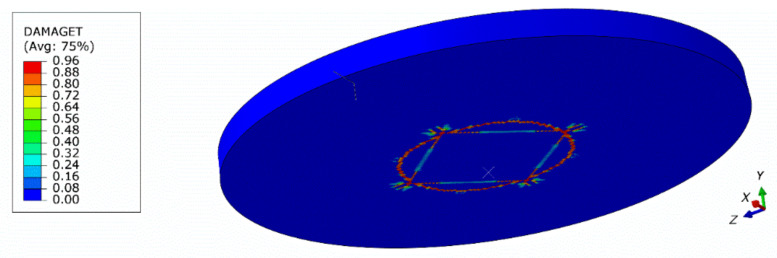
Contour plot of the tension damage parameter ($d_{t}$) after failure (3D model, global mesh size: 3 mm).

**Figure 19 materials-13-04034-f019:**
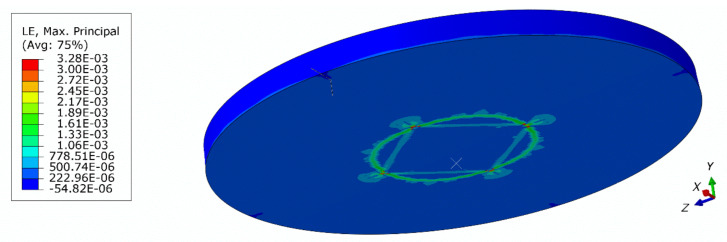
Contour plot of the maximum principal logarithmic measure of strain (3D model, global mesh size: 3 mm).

**Figure 20 materials-13-04034-f020:**
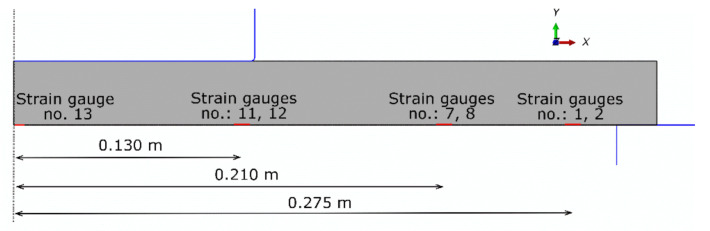
Locations of strain gauges used in the laboratory test presented in the axisymmetric model.

**Figure 21 materials-13-04034-f021:**
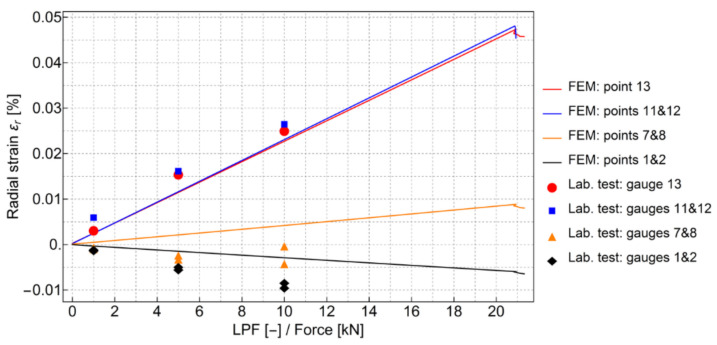
Comparison of radial strain between the experiment and finite element methodmodel (see also Figure 20).

**Table 1 materials-13-04034-t001:** The properties of vinyl ester resin used as the binder of tested mixes (www.ciechresins.com).

Property	Test Method	Value
Viscosity (25 °C)	DIN 53015	350 ± 50 mPa·s
Gelling time (25 °C)	ISO 2535	30 ± 5 min
Flexural strength	ISO 178	110 MPa
Tensile strength	ISO 527	75 MPa
Elasticity modulus	ISO 527	3500 MPa
Extension	ISO 527	2.8%
Heat deflection temperature	ISO 75	95 °C
Barcol hardness	ASTM D 2583	35 °B

**Table 2 materials-13-04034-t002:** Three-component curing system for vinyl ester resin used in tested mix (www.ciechresins.com).

Component	Function	Content (% of Resin Mass)
Cobalt naphthenate 1%	Accelerant	0.6
Dimethylaniline 10%	Accelerant	1.21
Benzoyl peroxide	Hardener	1.97

**Table 3 materials-13-04034-t003:** Mixing proportions by weight (kg) for a concrete volume of 1 m^3^.

Sand and Gravel	Resin	Fly Ash	Quartz Powder
1314 kg	329 kg	328.5 kg	328.5 kg

**Table 4 materials-13-04034-t004:** Material characterization results.

Test	Unit	Result	Standard Deviation
Compression (beam)	N/mm^2^	109.40	4.00
Compression (cube)	N/mm^2^	93.10	1.19
Compression (cylinder)	N/mm^2^	70.35	7.59
Bending	N/mm^2^	24.33	1.31
Tension	N/mm^2^	13.76	0.68
WST	N/m	17.68	0.39
Young’s modulus	kN/mm^2^	21.80	1.090

**Table 5 materials-13-04034-t005:** General and plasticity material parameters used for the concrete damaged plasticity model.

*ρ* [kg/m^3^]	*E* [GPa]	*ν* [–]	*K* [–]	*χ* [–]	*ψ* [°]	*f_b_*_0_/*f_c_*_0_ [–]	*μ* [s]
2400	21.802	0.2	0.667	0.2	36	1.05	0.005

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
