# Peer review of "Prediction of Structural Performance of Vinyl Ester Polymer Concrete Using FEM Elasto-Plastic Model"

_materials, 2020, doi:10.3390/ma13184034_

Round 1

Reviewer 1 Report

The study ”Prediction of structural performance of vinyl-ester polymer concrete using FEM elasto-plastic model” analyzes the mechanical properties of vinyl-ester polymer concrete using a FEM method. The analyzed material implies recycling waste products that give to the concrete an ecologic character and by that the authors follow the general trend in the ”battle” for a cleaner environment and sustainable building materials.

The Abstract presents, in short, the subject of the developed research, it covers the motivation and the methods used in the study, It needs to be improved with a short presentation of the results obtained.

In the Introduction section, the paragraphs from the 50-72 rows must be moved before the 41 row for more fluency of the text.

The section of Mix Design must be improved with more details about the statistical design method. Please provide the mixes prepared for testing, with their components and quantities or percent, in a separate table, not in common with the results of the compressive or tensile strength.

Figures from the 2 and 3 sections that imply results must be moved in the Results section and discussed there.

The authors must provide a general explanation of the principle or steps of the mathematical design and analysis method, at the beginning of the Methods section. It is not clear why it is started with a Box compositional design (row 131, 2.1.4 Mix design), and then it is a jump to 3. Development of the finite element model. Some details between them are missing.

Please structure the article according to a standard research article into Introduction, Materials & Methods, Results & Discussions, and Conclusions as main chapters, then in other subchapters according to your study (I refer mainly to section 3 that starts in 194 row). In the Materials and Methods section must be found only descriptions, not results also.

It lacks the Conclusion section, where the authors must synthesize the obtained results of the study.

English must be enhanced as well.

Reviewer 2 Report

This paper deals with constitutive laws for polymer concrete (PC) to be used in FEM analysis. The obtained constitutive laws are employed to analyze the PC manhole covers. The obtained results are compared with the experimental results. The structure of this paper is clear. However, it is difficult to understand the originality of this paper because the results obtained by the experiment and the preinstalled conditions of the analysis software (Abaqus) look mixed. The CDP may be pre-implemented in the Abaqus software. The reviewers have never used the Abaqus software, so I have no idea about its detail. The authors should highlight the new findings so that the originality of the paper is clear. In addition, unlike the typical paper format, there is no concluding chapter at the end. The lack of conclusion chapter also makes the originality of the paper difficult to understand.
Please consider modifying the paper based on the following comments

・ There are two eq. (2) in L. 222 and L. 235. Correct the equation numbers appropriately.

・ Table 1 shows only the physical properties of the vinyl-ester resin. Please indicate chemical composition as well, If possible.

・ In 2.1.2, “Aggregate material sieved into fractions 0.125/2mm and 2/4mm was used to prepare tested concrete mixes (1/3 to 2/3 in weight).” This sentence is difficult to understand, especially meaning in the bracket. Did the authors use 0.125/2 mm for 1/3 of the total aggregate weight and 2/4 mm for 2/3 of the total aggregate weight, respectively? A graph of the size distribution after the sieve may make readers understand easily.

・ Regarding 2.1.3 Microfiller, the microfiller effect in cement concrete is obtained by powders that are sufficiently smaller than the cement particles, for example, silica fume. Is the fly ash used here as a filler sufficiently smaller than the particle size of the resin? In this case, the particle size of the resin should be indicated. Measured size distributions are also required for fly ash and quartz powder.

・ Figure 1 shows a graph that calculates compressive and tensile strength using A/B and P/M as parameters. However, the mix proportion shown in Table 3 is one of these points. First, please explain how these graphs were obtained. Next, please indicate where the values in Table 3 are located in these figures. In L. 140, B/M is defined. Was this value also used in the graphs or the paper?

・ Table 4 is not explained in the body text. Please add appropriate descriptions.

・ Regarding 3.1, please reconsider the description of this paper’s originality, as I pointed out at the beginning of this review comment. The adoption of CDP may be one of the most important suggestions. For example, the values shown in Table 5 might be obtained from the results of experiments shown in Table 4. In the text, the authors say, “Herein, all these parameters were assumed based on laboratory test results, literature or Abaqus suggested default values.” To make the originality of the paper clearer, new findings of this paper should be highlighted.

・ I have no idea how to understand Figure 10. Figure 10a looks like just a piece of a graph (square) paper. Figure 10b is unclear, either. Only the yellow area is described as having an aspect ratio higher than 1.5. However, the resolution of the figure is insufficient, and I cannot see how the mesh is divided. Please replace it with more exact pictures.

・ Unlike the typical paper format, there is no conclusion chapter at the end. Please add a conclusion/summary chapter.

Reviewer 3 Report

Comments

This study experimentally and numerically investigated the performance of vinyl-ester polymer concrete. The findings of this study are interesting and add value to the knowledge. The paper can be considered for publication after addressing the following major revisions:

  • The author need to highlight the applications of polymer concrete in introduction section. Recently polymer concrete was applied for manufacturing railway sleepers and structural beams [Ref: Evaluation of an innovative composite railway sleeper for a narrow-gauge track under static load, and Flexural and shear behaviour of layered sandwich beams]. Suggest to include this information.
  • Does the author tested the properties given in Table 1 and Table 2? If not, then please provide the reference.
  • The grading curve of the aggregates should be provided in Section 2.1.2 as the properties of the concrete largely depends on it.
  • What chemical compounds are there in the fly ash and quartz powder?
  • Table 3 shows only one mix ratio. How Fig. 1 was plotted with one mix ratio?
  • How tensile strain was measured in Fig. 3? Please clarify.
  • Suggest to present stress-strain curve in Fig. 5 instead of load-displacement.
  • How polymer concrete manhole cover can be a suitable replacement for cast-iron manhole cover? Does it have enough strength to carry wheel load of the vehicles?
  • This paper discuss the modelling of polymer concrete. The reviewer believe that the author would be benefited from the modelling of polymer concrete presented in the paper titled Optimal design for epoxy polymer concrete based on mechanical properties and durability aspects. Suggest to include the key findings.
  • The author need to present the key findings in a conclusion section.

Round 2

Reviewer 1 Report

Due to the fact that the authors meet all of the requests from the first reviewing report, I agree with the article publishing as it is now.

Author Response

We wish to express our appreciation for your in-depth comments, suggestions, and corrections, which have greatly improved the manuscript.

Reviewer 2 Report

The authors have well modified the manuscript according to the reviewers' comments except for the 4th comment of Reviewer 2. This journal deals with various kinds of materials and is not intended only for polymer cement concrete specialists. Please add proper explanations of the microfiller effect and its mechanism in polymer cement concrete, as the authors described in the cover letter.

Author Response

We wish to express our appreciation for your in-depth comments, suggestions, and corrections, which have greatly improved the manuscript.
Regarding round 2 of the reviews - according to the request (comment 4th) we added proper explanation of the microfiller effect and its mechanism in polymer cement concrete (section "2.1.3. Microfiller") as it was described in the cover letter. We believe that the revised version can meet the journal publication requirements.

Reviewer 3 Report

I have no further comments.

Author Response

(The authors gave the same response as above.)
